# Case Study in a Working Environment Highlighting the Divergence between Sound Level and Workers’ Perception towards Noise

**DOI:** 10.3390/ijerph17176122

**Published:** 2020-08-23

**Authors:** Chun-Yip Hon, Illia Tchernikov, Craig Fairclough, Alberto Behar

**Affiliations:** 1School of Occupational and Public Health, Ryerson University, Toronto, ON M5B 2K3, Canada; 2Workplace Safety and Prevention Services, Mississauga, ON L4W 0A1, Canada,; Illia.Tchernikov@wsps.ca (I.T.); Craig.Fairclough@wsps.ca (C.F.); 3SMART Lab., Ryerson University, Toronto, ON M5B 2K3, Canada; albehar31@gmail.com

**Keywords:** occupational noise, attitudes and perceptions, mixed-methods, meat processing, hearing conservation program

## Abstract

Excessive noise levels are a prevalent issue in food processing operations and, although there have been numerous studies on occupational noise, no single study has used a concurrent mixed-methods approach. Employing this study design allows for an understanding of the level of convergence (similarity) between measured noise levels and workers’ attitudes and perceptions towards noise. This, in turn, allows for the identification of potential challenges with respect to the implementation of hearing conservation efforts. In this study, spot noise measurements were collected using a sound level meter. One-on-one interviews were conducted with workers to determine attitudes and perceptions towards noise in their workplaces. Subsequently, the results of the noise measurements (quantitative data) were integrated with the survey responses (qualitative data) to identify convergence. The majority of the noise measurements were found to exceed 85 dBA—the criterion mandated by the local occupational health and safety legislation. Although all study participants felt that it was noisy in the workplace, a large proportion of respondents indicated that the noise was not bothersome. With workers’ perception being contradictory to the measured noise levels, it is a challenge to implement hearing conservation measures unless changes are made to raise the awareness of the risks associated with excessive noise exposure.

## 1. Introduction

Noise is a common occupational hazard found in many workplaces across the globe [1,2]. High noise levels within food processing facilities are no exception as a variety of equipment and activities including, but not limited to, cutting, grinding, mixing, bottling/canning, and washing are known to be noisy [3]. Studies have found that noise levels in the food processing sector routinely exceed 85 dBA [2,4] which is the typical occupational exposure limit adopted by many industrialized nations [2].

Excessive noise exposure can lead to fatigue, distraction, and difficulty communicating—all of which can result in workplace injuries [5]. From a health perspective, excessive noise is associated with cardiovascular effects [6] as well as noise-induced hearing loss [7]. In fact, noise-induced hearing loss is the number one disease claim from the food processing sector in the province of Ontario, Canada (data obtained through an electronic request to Ontario’s Workplace Safety and Insurance Board).

Fortunately, excessive noise in the workplace is preventable [8]. Studies have indicated that it can be reduced through the implementation of a hearing conservation program [9,10] which typically includes noise exposure assessments, noise control measures, audiometric testing as well as worker training [11,12]. However, prior to the implementation of such a program, it is important to assess the likelihood that workers will engage in such a program by understanding their attitudes and perceptions towards occupational noise [13,14]. To the best of our knowledge, no single study has examined this common occupational issue using a mixed-methods approach, which involves collecting noise measurements and concurrently ascertaining the attitudes and perceptions of exposed workers through qualitative analysis. Therefore, the objective of this novel study was to establish if there is any convergence (similarity) with the two types of data and, in turn, ascertain potential challenges with the introduction of hearing conservation efforts in food processing facilities. For this study, convergence was defined as the noise measurement being ≥85 dBA and the corresponding worker perception was that the environment was considered noisy. This study was conducted exclusively within meat processing operations, where noise sources in these facilities include noisy equipment/procedures such as deboning, slaughtering, grinding and pulverizing [3]. Other types of food processing operations such as baking, beverage, or produce were excluded as they could potentially confound the results given that those operations include different noise sources. This was an exploratory study as noise has previously been found to be excessive in food processing facilities [4,15]); however, it is unclear if workers’ perception of their risk is likely to negatively impact hearing conservation efforts in these workplaces.

## 2. Materials and Methods

This study was approved by institutional research ethics prior to its start (Ryerson REB# 2015-291-1). The exploratory study was part of a larger ongoing project examining occupational health and safety issues within the meat processing sector in Ontario, Canada which is the most common type of manufacturer in the province’s food and beverage processing industry [16].

### 2.1. Site Recruitment

Electronic invitations to participate in the study were sent to a database of meat processing facilities that had employees who were former graduates of Ryerson University’s School of Occupational Health and Public Health (“School”) or who had completed occupational health and safety courses offered by the School. The inclusion criteria were that the target organization was primarily involved with meat processing and had a facility located in the Greater Toronto Area.

If the representative from the organization agreed to participate, an ethics-approved study information sheet was provided via email. The information sheet was subsequently distributed and/or posted to workers at each site to: (a) explain the scope of the study and (b) outline the expectations of individuals who may be approached to be interviewed (see below).

### 2.2. Interviews

Worker interviews were conducted on site visit days, mutually agreed upon between the research team and the participating food processing facility, i.e., convenience sampling (a type of non-probability sampling method where the sample is taken from a group of people easy to contact or to reach (from dictionary.com)). At least one worker from the different areas of the plant where noise measurements were collected was approached to participate. Since each interview took up to 45 min, the maximum number of interviews per site was six as the research team was scheduled to be at each facility for approximately five hours. Exclusion criteria included those individuals who did not speak English or who had difficulties in understanding terminology such as noise-induced hearing loss, noise control measures, and hearing protective devices. After verbally agreeing to participate in the interview, individuals were given a copy of the consent form to review and sign prior to the one-on-one interviews.

The interview questions were adapted from a questionnaire developed by Davies and Shoveller [17] which has been used previously to examine noise in food manufacturing [11]. The sections of the aforementioned questionnaire employed in the current study were: (1) demographics, (2) workers’ perception and attitude of noise, and (3) perceived challenges in reducing noise exposure. One member of the research team (CYH) conducted all the interviews and every interview was audio recorded for subsequent transcription.

Once transcribed, the responses were analyzed via content analysis whereby similar comments were grouped together under one category. Research team members independently grouped the interview responses and met afterward to review the groupings until consensus was reached. Subsequently, the frequency of responses was tallied and select comments, which encapsulated the views of the respondents, were extracted. This approach was similar to the one used by Hong et al. [18] to assess the perceptions and attitudes of firefighters to noise.

### 2.3. Noise Level Measurements

An initial walk-through of each site was conducted to review the layout, examine the different operations and identify locations where area noise measurements could be performed without obstructing workers and/or traffic (foot or motorized). Noise measurements were collected on the same day as the worker interviews and were intended to qualify the noise levels.

Two Larson Davis SoundTrack LxT integrating sound level meters (SLM) (Depew, NY, USA) were employed to collect noise measurements at each participating site. The SLMs were set to the “A” frequency weighting and “slow” for the detector. Prior to sample collection at each site, SLMs was calibrated using the Larson Davis CAL150 Acoustic Calibrator (Depew, NY, USA).

At each site, noise measurements were collected in production and packaging areas during normal daily operations. The SLMs were used to collect spot measurements around each noise source. Five to 14 pairs of measurements were obtained in every department assessed (the number of measurements was dependent on the physical dimension and layout of each department). The instruments were placed on tripods at approximately 1.5 m from the ground in accordance with standard occupational hygiene practices. Sound levels were measured at 1 m increments from each machine or noise source (maximum of 3 m from each), which is representative of where a worker may be situated, and were collected for at least 2 min each in accordance with Canadian Standards Association (CSA) Z107.56-13—“Measurement of noise exposure” standard [19].

As per the CSA Standard on Hearing Loss Prevention Program Management [20], the first step is performing a noise level survey (spot measurements) to identify potential hazard areas and noise sources. If the results exceed 85 dBA, then a noise exposure survey has to be performed to quantify the hazard. Subsequently, if the full shift noise exposure levels exceed 85 dBA, then a hearing conservation program has to be implemented. Therefore, in the present study, 85 dBA was selected as the “safe” sound level limit.

### 2.4. Data Convergence

To ascertain the extent to which the two sets of data converged [21], the results of the interview data (qualitative statements) were integrated with the noise level measurements (quantitative data). For the purposes of this study, convergence occurred when the spot noise measurement was ≥85 dBA and the corresponding worker perception was that the environment was noisy.

## 3. Results

None of the participating sites had a complete and thorough hearing conservation program in place. Each had selected elements of best practices such as warning signs, non-routine audiometric testing, etc. which varied between sites. Although hearing protective devices (HPD) were made available to all workers at every site, their use was irregular with most workers wearing the devices for part of the work shift.

### 3.1. Interview Responses

#### 3.1.1. Respondent Characteristics

Every individual who was approached to be interviewed agreed to participate and all met the inclusion criteria (100% response rate). Overall, 22 individuals across all four sites participated in one-on-one interviews. The majority of respondents were female, middle-aged, had 5–14 years of experience in their current job and possessed a high school diploma as their terminal level of education (Table 1). None of the respondents reported that they had been medically diagnosed with noise-induced hearing loss.

#### 3.1.2. Attitude and Perception of Noise in the Workplace

All 22 respondents (100%) felt that it was noisy every day and, according to 82% of those interviewed, it was noisy for most of the shift (not shown). Regardless, only 54.5% of respondents were concerned about noise in the workplace. In fact, 13.6% of respondents felt that noise, in general, is not an important occupational issue and 22.7% of workers did not feel that this matter deserved any further attention, i.e., the status quo was acceptable to them. Moreover, nearly 10% of respondents did not agree with the statement “you would feel better if your workplace was quieter”. However, a majority of respondents (90.9%) did indicate that more education/awareness was needed with respect to occupational noise (Table 2).

#### 3.1.3. Perceived Challenges in Reducing Noise Levels

When interviewees were asked the question “What do you think are the best solutions to noise problems in your workplace?” several respondents felt that it was not feasible. Select responses from the interviews include:

“I don’t think they can do anything to reduce noise”

“No solution”

“No idea”

In fact, according to one respondent, it would take “a miracle” to reduce the noise levels. Overall, many respondents felt that personal protective equipment, specifically HPD, was sufficient to reduce their risk of exposure.

When asked, “What kinds of problems would make it difficult for a new noise control measure to be developed and implemented?” there was one predominant theme—ensuring that the novel control measure meets food quality/safety standards. Some of the responses to this question were:

“Any changes in health and safety needs to also fall within the food safety requirements. Usually a solution that can work for health and safety may not be friendly for the food safety side. For example, equipment change can create harborage sites where bacteria can build up, which is not suitable for food safety”.

“Food quality is top priority”

“Need to be careful regarding the sanitation aspect of it.”

#### 3.1.4. Data Convergence—Noise measurements vs. Interview Responses

Figure 1 displays the outcome of the data convergence by plotting the measured noise levels with the corresponding category of responses from the interviews (one of: “it is very noisy”, “neutral” or “not noisy at all”). Overall, there was a divergence in almost 50% of the participants where a worker indicated that the noise was not an issue for them even though the spot measurement result exceeded 85 dBA (or vice versa). Table 3 lists the noise measurements with the corresponding worker perceptions and a number of contradictions can be seen. The following are examples of discordant statements made by respondents:

“It’s not that bad. The noise doesn’t bother me”—Worker 3E from Site 3 where the average noise levels exceeded 85 dBA.

“I’m okay with noise in the workplace”—Worker 4E from Site 4 where the average noise levels exceeded 85 dBA.

This discrepancy between excessive sound measurements and worker indifference to noise was most notable from Sites 3 and 4.

## 4. Discussion

By employing a mixed-methods approach, this exploratory study identified instances of disagreement between the attitudes and perceptions of workers regarding occupational noise and the measured sound levels in the workplace. In essence, despite the fact that the majority of the spot noise measurements exceeded 85 dBA, about half of the respondents were cavalier about noise in their workplace. Specifically, these indifferent workers stated that the noise was not an issue and/or they were not bothered by it. With such a large proportion of exposed workers being aloof, this presents a significant barrier with respect to implementing hearing conservation efforts as disenchanted workers can negatively affect the success of occupational health and safety initiatives [22,23]. These results also support the assertion that workers do not fully comprehend the risk that they are exposed to [24] and that workers are becoming “normalized” to the excessive noise levels [25]. To overcome this, Hétu suggests that raising awareness of this occupational issue is vital and that those individuals who are experiencing some form of hearing loss disclose their condition in order to spearhead change [26]. In addition to a change in attitude and perception, there also needs to be an overall cultural shift with respect to health and safety in the workplace [27]. Essentially, an effective approach should be considered to improve the organization’s commitment to reducing the risk of noise exposure [28,29].

Another issue that could be a hindrance to the successful introduction of a hearing conservation program is that many of the participants were resigned to the fact that noise is “part of the job” and believed that the use of HPD is sufficient to the risk. It has also been argued that where workers deem the workplace as not being a noise risk, then they will believe that it is not necessary to employ protective measures [24].

This false notion that reliance on HPD could save workers’ hearing was also reported in a Swedish study [30]. Furthermore, respondents stated that HPD was not always worn in loud circumstances even though the employer made the devices available (not shown). This is not a novel finding as poor compliance with HPD usage in the manufacturing sector have been reported by others [9,25,30]. If HPD are provided by the employer to protect workers’ hearing, then management must improve its efforts with respect to enforcement of their use [31,32]

In addition to the use of HPD, an effective hearing conservation program should also include engineering controls [13,33] but, since these types of controls require more initial capital [34], there may be substantial cost considerations. However, what is more of an issue, as alluded to by several study participants, is the need to ensure that the engineering control can be cleaned/sanitized in order to meet food safety requirements. Health and safety interests competing with non-safety-related requirements is an ongoing issue in the food manufacturing sector [35]. However, this limitation should not preclude noise control considerations such as enclosing and/or isolating noisy machinery. Therefore, every effort should be made to consider these control measures to reduce the risk of noise exposure in these facilities.

It bears mentioning that this study was timely as the province of Ontario had, at the time that sample collection took place, recently updated the occupational noise regulation [36]. The current study found that most participating facilities were not in compliance with many of the requirements of this newly-introduced legislation including the occupational exposure limit for noise. In fact, the maximum recorded noise level of 141.6 dBA is of concern as the National Institute for Occupational Safety and Health (NIOSH) has indicated that the threshold of pain can be as low as 120 dB [37]. Adhering to the minimum legislated criteria is necessary in order to reduce the risk of excessive noise exposure and, subsequently, to implement hearing conservation programs.

This study is only representative of the participating sites and the times when sampling took place; noise levels may vary at other sites and at other times. Nevertheless, as indicated earlier, excessive noise is a common occurrence in food processing and the noise measurements were collected during normal operations. Although the sample size was limited, excessive noise was found in most participating sites and divergent results were reported in all four facilities. However, the divergent results could be attributed, in part, to an increased hearing threshold of workers due to hearing loss [38,39]). As such, it is recommended that future studies consider including more sites (including other types of workplaces), a larger number of participants as well as audiometric testing of interview respondents. Although the perception results in the current study are likely not unique to the meat processing industry [24,25]; due to the study design, we do express caution about the generalizability of the results. Finally, the interview responses can only be attributed to each individual respondent and cannot be generalized to other workers within the same department/facility.

## 5. Conclusions

In summary, this exploratory study confirmed that excessive noise (sound levels greater than 85 dBA) is prevalent in the participating facilities; yet nearly 50% of the exposed workers were nonchalant about this occupational hazard even though they recognized that the environment was loud. In addition, the results uncovered a fatalist attitude amongst many workers as well as their mistaken belief that HPD alone is sufficient to reduce the risk. Combined, this makes hearing conservation efforts within the meat processing industry challenging as well as necessary. Modifying worker attitudes towards noise by raising awareness of the risk associated would be an important first step to the implementation of hearing conservation efforts.

## Figures and Tables

**Figure 1 ijerph-17-06122-f001:**
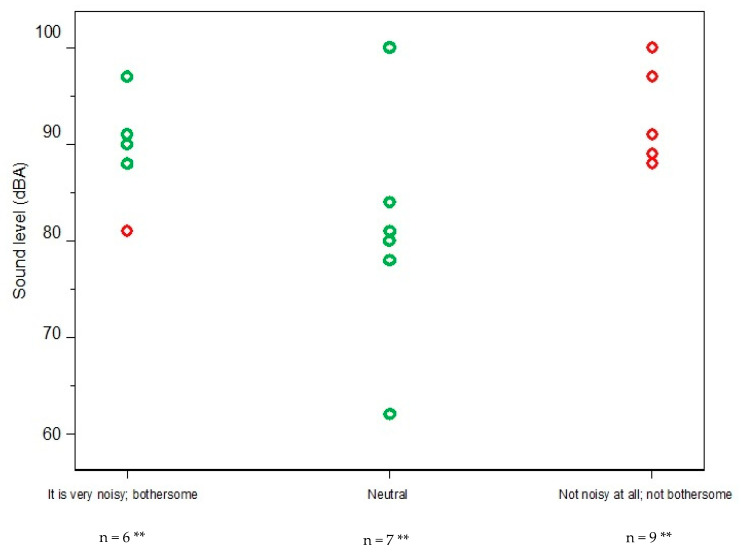
Plot of noise measurements * and corresponding worker perceptions where green represents convergent results and red represents divergent results.* the sound levels are the averages of the spot noise measurements from the worker’s department(s) ** some of the sound values overlap and therefore do not appear as unique points in the plot.

**Table 1 ijerph-17-06122-t001:** Characteristics of respondents (N = 22).

Variable	Subcategory	%
Sex	Female	54.5
	Male	45.5
Age	25–34 years old	18.2
	35–44 years old	27.3
	45–54 years old	27.3
	55–65 years old	22.7
	65+ years old	4.5
Highest education level	High school or less	59.1
	College	22.7
	University	18.2
Tenure in current position	<1 year	9.1
	1–4 years	13.6
	5–9 years	27.3
	10–14 years	27.3
	15–19 years	4.5
	20+ years	18.2

**Table 2 ijerph-17-06122-t002:** Attitudes of workers regarding occupational noise. (N = 22).

Question	Yes/Agree (%)	No/Disagree (%)	Don’t Know (%)
Noise is an important occupational health and safety issue	86.4	13.6	-
Noise should be given more focus/attention/discussion	77.3	22.7	-
Noise is an issue not always taken seriously	54.5	45.5	-
People need education/awareness regarding noise	90.9	9.1	-
You would feel better if your workplace was quieter	77.3	9.1	13.6
You worry about noise in your workplace	54.5	31.8	13.6

**Table 3 ijerph-17-06122-t003:** Workers’ perception of noise vs. measured noise levels. Divergent results are italicized. Divergence = noise measurement was ≥85 dBA and the corresponding worker perception was that it was not noisy (or vice versa).

Site. (Number of Interviewees)	Worker Identification	Department(s)	Range of Measured Noise Levels (Average) (dBA)	Perception of Noise Verbatim Comments [Perception Category *]
1—Cured meats (n = 4)	Worker 1A	Ready to Eat	88.2–111.0 (100)	*I don’t have no problem, you know. I don’t, think you need to wear the things (ear plugs) you know, that’s what I say* [A]
	Worker 1B	Ready to Eat	88.2–111.0 (100)	It’s always the same noise. I don’t think nothing can be done, because it’s the machines [B]
	Worker 1C	Kitchen/raw preparation	85.1–109.2 (97)	It’s pretty loud. They provide these (ear plugs) and it’s just a matter of people wearing them. They are the best solution to the noise problem [C]
	Worker 1D	Kitchen/raw preparation	85.1–109.2 (97)	*The machines that I use, they are not really that noisy* [A]
2—Meat seasoning (n = 6)	Worker 2A	Bakery	82.6–84.8 (84)	8 h exposure every day, better now than it once was in my area (bakery). Still 80 to 85dB. There’s a lot of vibration and everything, which is also very tiring and wearing on the body [B]
	Worker 2B	Bakery; Bagging area; Maintenance; Mixing floor	82.6–84.8; 82.3–84.5; 75.7–84.1; 70.8–78.3 (81)	*It’s definitely loud and over 85. Like you’re yelling at one another* [C]
	Worker 2C	Bakery; Maintenance; Mixing Floor	82.6–84.8; 75.7–84.1; 70.8–78.3 (80)	The company is adequate in trying to protect your hearing. My exposure changes day to day too. Noise is effectively controlled by the employer [B]
	Worker 2D	Bakery; Mixing Floor	82.6–84.8; 70.8–78.3 (80)	Not uncomfortably loud, but loud. I can tolerate the noise, as it’s at the threshold. [B]
	Worker 2E	Production	77.3–84.1 (81)	The noise level is definitely on the higher side; but at the threshold for the maximum allowed [B]
	Worker 2F	Production	77.3–84.1 (81)	*Especially when all the machines (sic) going, it’s really loud noise. It’s really annoying. I have an appointment with my doctor next month, as I am experiencing hearing loss. There was no problem before I started* [C]
3—Cured meats (n = 6)	Worker 3A	Slicing; Packaging	85.3–87.6; 84.7–90.6 (88)	Both areas where I work are noisy [C]
	Worker 3B	Raw Production	88.5–92.4 (91)	*It’s loud, especially when the machines gave (sic) that banging, when it gets ready to finish. I’m so use to not wearing them (ear plugs), you get comfortable with the noise* [A]
	Worker 3C	Raw Production	88.5–92.4 (91)	*It’s (the noise) usually the same from when you start until when you end. I’ve always worked in loud environments - never bothered me. There’s always ear plugs, so if it’s too noisy you wear ear plugs right?* [A]
	Worker 3D	Raw Production	88.5–92.4 (91)	*Not affected by it—I can tolerate it* [A]
	Worker 3E	Packaging	84.7–90.6 (88)	*It’s not that bad. The noise doesn’t bother me* [A]
	Worker 3F	Maintenance	72.6–82.1 (78)	It (the noise) is bothersome of course for anyone—I’m not too sensitive to it. [B]
4—Deli meats (n = 6)	Worker 4A	Packaging	89.9–91.7 (91)	Personally, the noise bothers me. Sometimes I get headaches, because we work in there so long, we get used to it. [C]
	Worker 4B	Maintenance	47.5–75.4 (62)	It’s not that bad, because we get training, if that machine is going to make noise we have to wear protection, right? [B]
	Worker 4C	Packaging	89.9–91.7 (91)	*Really, sometimes it bother me, sometimes it don’t. You know. Because when I’m working, I don’t really pay attention. Sometimes I tune it out and sometimes I wear earplugs.* [A]
	Worker 4D	Ready to eat	85.5–91.5 (89)	*For me, it’s not too noisy. Even I can work without ear plugs* [A]
	Worker 4E	Packaging	89.9–91.7 (91)	*I’m okay with noise in the workplace.* [A]
	Worker 4F	Packaging; Ready to eat	89.9–91.7; 85.5–91.5 (90)	In some areas, it can be unbearable. I personally do carry earplugs with me and when I find it unbearable I will put them on [C]

* Perception categories: [A] = Not noisy at all, not bothersome; [B] = neutral; [C] = It is noisy, bothersome.

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
