# Peer review of "Case Study in a Working Environment Highlighting the Divergence between Sound Level and Workers’ Perception towards Noise"

_ijerph, 2020, doi:10.3390/ijerph17176122_

Round 1
Reviewer 1 Report
Dear editor, dear authors;
This paper focuses on analyzing the perception of workers in meat processing facilities about environmental noise in their workplaces. The authors argue that if there is a "convergence" between noise levels above 85 dBA and a negative perception of noise by workers occupying these spaces, it may be the basis for facilitating future hearing conservation programs. Only certain processing operations are under the scope of the study. For this, 22 people were chosen to be interviewed in 4 different facilities. For the on-site noise measurement campaign, 2 sound level meters were used. For this purpose, 5 to 14 pairs of measurements were obtained in each assessed department. Table 3 and Figure 1 group the results giving a discouraging view of the possible correlation between the increase in noise and an increasingly negative response from the workers.
In summary, the topic is suitable for the journal and the title of the paper anticipates what the article is about. Maybe, in my opinion, the title is too long. The justification of the study and the significance of the work is clear since the subject under study is important. It is also well written, and easy to read. The abstract is meaningful and provides an accurate representation of the article. The extension of the manuscript is satisfactory given the contents addressed. The structure of the paper is correct and the contents of the study are clearly exposed. The scientific environment in which the research takes place is introduced correctly with some shortcomings that will be discussed below. The introduction to burning issues is very clear and well written. The explanations are easy to follow. Unfortunately, and in my opinion, the references supporting the introduction have two weaknesses. (i) The references cited in the text are not current (only 5 of the 32 references were written in the last five years). Furthermore, (ii) the revision of the role of human perception towards noise is not of the importance required by the study and, on the contrary, the "non-scientific" citations are excessively important. However, the reader realizes that the research is not isolated and is linked to other existing works
The interpretations and conclusions are justified by the data, but within certain limits, which I think the author should discuss and try to justify in greater depth. Another positive issue is that this paper includes reasons that justify raising awareness of noise transferable to managers and workers of the meat industry.
For that reason and after a careful reading of the paper:
IJERPH-889664, Titled: Convergence of noise measurements with workers’ attitudes and perceptions towards noise: Understanding their collective impact on hearing conservation efforts which the first Author is Chun-Yip Hon
my recommendation is to publish it with some major corrections.
Throughout this letter, I will try to explain the reason for this evaluation, trying to highlight the areas for improvement.
MAIN CONCERNS
I am examining a prospective and unfinished study (in the Conclusions, the authors define their study as "exploratory"). In my opinion, after the first round of measurements and interviews, the authors should have asked themselves more questions that would have led them to collect more data and, perhaps, other variables that could give a deeper explanation of the first results found. Overall, this would provide a broader analysis that would translate into a deeper and more robust discussion that would provide readers with more insight. This means the need for the authors to review the state of art and the methodology. Regardless of the interest of the study, in its current version, it is estimated that the degree of reliability and generalization of the conclusions is limited.
It is widely known that noise in the workplace, if excessive, is negative for health, regardless of its origin. Both auditory and non-auditory effects of noise on health are found in the literature. However, this is a study on workers' perception of noise. A considerable number of works available in the bibliography has given us the guideline to understand that there is a high psychological component in this perception of acoustic problems.
Concerns and areas for improvement
(i) After reading this study, it can be deduced that the authors handle the (implicit) hypothesis that workers in meat factories suspect the ability to reduce noise without altering the sanitary quality of the products they manufacture. In this situation, it seems that workers will tend to defend their jobs. Consciously or unconsciously, they may assume that enduring noise may be "the least bad option."
To prove this statistically, my question to the authors is the following:
don't you think it would be convenient to carry out a study with similar characteristics (number of interviewees, number of factories, the noise level in the workplace) in the same region of Ontario (to avoid local factors such as legislation, business culture, social factors, etc.), but in different types of noisy factories?
(ii) It seems necessary that the authors should include in the introduction bibliographic references that help to understand the psychological connotations of noise perception in the workplace. Following this argument, my question to the authors is the following:
Shouldn't they discuss and justify if they should incorporate in the study the measurement of other types of variables (acoustic and non-acoustic) for the correct interpretation of the relationships between the noise level and their perception?
For example and talking about the need to include a more extensive description of the acoustic variables collected:
Why do not use the individual noise dose for a daily evaluation exposure to a high average noise level? Perhaps the dose explains better the results of interviews. Why are noise logging statistics not been included during the measurement time?
(iii) As the authors acknowledge: "Although the sample size was limited, excessive noise was found at most participating sites and divergent results were reported at all four facilities." Following this recognition of the limits of their work, I want to ask the authors the following questions.
Do the authors not believe that they should correct this situation to provide statistical significance to their statements (for example, that almost 50% of exposed workers are indifferent to this type of occupational risk)? Would it not be advisable to increase the number of cases?
OTHER MINOR CONCERNS
- Is there a difference between the four factories? Why not highlight in figure 1, distinguishing the results from different factories?
- Overlapping the results in Figure 1 should be avoided, and therefore avoid appearing as single points on the graph.
- About the workers. Why have workers not been differentiated by the time they have held jobs positions subject to high noise levels?
- For clarity in the presentation, it would be interesting to create an annex with the set of optional questions and answers that have been used in this document.
- Please check some minor errors in the text. Example: Line 229. You may consider replacing "If HPD is provided…" with " If HPDs are provided…" or “If HPD is provided..”
- In your experience, could you affirm that some characteristics of noise (tonality, impulsivity, etc.) are not interesting to evaluate the perception of noise nuisance in the workplace?
Author Response
Thank you for taking the time to review our manuscript. Please see the attachment.

Reviewer 2 Report
The manuscript reports findings of noise measurements and interviews with staff at meat processing sites in Canada. It found that noise levels were generally high, and that worker attitudes towards noise were mixed. I thought the work was good but didn’t like the way it was presented.
Overall the research methods do not seem appropriate for the stated aims. From the title (and elsewhere) the aim was to assess the “convergence” between the measured noise levels and the reports of workers, however the methods don’t really support this. The idea of seeking correspondence between judgements of noise levels and measured noise levels is interesting, but it would need to be done on a more precise basis: dosimetry, more careful questioning, relating the judgements to specific noise levels rather than generalised reports. Figure 1 shows a null result, and in principle I like to see null results published, but in this case it isn’t an interesting finding because of the lack of control of the measurements on which it is based. But I don’t mean to seem negative though: the work was still good and I suggest that removing the claim that the research was conducted in order to investigate convergence would avoid the issue about the methods and allow the work to focus on the other strengths, which would be worth presenting.
The sound level measurements were taken with stationary meters according to standard protocols, and the findings are not unexpected: they are generally in the range of around 80-100 dB with the exception of one measurement, that was about 60 dB: it would be helpful if the authors could explain why that particular measurement was anomalous. Essentially, the measurements confirm that noise levels were generally high and this is useful background to the main part of the work, the interviews.
The interviews were 45 minutes long and must have collected a great deal of data, though little of this is presented. There is a big table that presents a single quote from each participant, alongside a brief description of the person and the sound levels where they work. If the authors were to focus the manuscript on the interview data and perform a simple but comprehensive general thematic analysis, I suspect there would be very interesting findings. Previous work (such as Reddy et al., 2014, which the authors cite) has done a similar thing, but this kind of thing is rarely published, and it would make for an interesting comparison.
A few minor points:
Line 23: even though you have used inverted commas, it isn’t really right to say “safe” for a sound level of 85 dB. Better to say “criterion mandated in national OSH regulations” or something similar.
Line 32: noise is not present in every workplace. Maybe ‘many workplaces’?
Line 49: “mix-methods” should be “mixed-methods”
Author Response

(The authors gave the same response as above.)

Reviewer 3 Report
In this manuscript the authors present the convergence of noise measurements with workers ’attitudes and perceptions towards noise, these are some of my comments
A weakness of the study is that the antecedents of the interviewees are unknown since they were contacted electronically, these antecedents must be specified.
Selecting a member from the different areas complicates the investigation since it makes the sample very heterogeneous.
The sample size is small.
It is important to define the hearing capacity of the participants.
Was any hearing impairment ruled out in the participants?
The sample is very heterogeneous in terms of age, type of occupation and time spent in the area.
Some variables such as place of residence, previous noise exposure and work history of similar situations should be considered.
The main concern is that the study presents a very small and heterogeneous sample and does not show a series of important variables that influence the study result.
Author Response

(The authors gave the same response as above.)

Round 2
Reviewer 1 Report
Dear editor, dear authors;
This is the second revision of a paper that focuses on analyzing the perception and attitudes of workers in meat processing facilities about environmental noise in their workplaces.
This reviewer identified some parts of the work that should be reviewed and, where appropriate, corrected, or answered by the authors. It is clear from the authors' response that there are certain aspects of my first review that cannot be corrected and will in any case need to be addressed in future work. Therefore, the study remains practically the same after some small corrections, and it should be considered "exploratory" and admit that the conclusions are not generalizable and should be judged assuming the limitations inherent in the study design.
Even under these circumstances, there is something that still worries me. As I said earlier in my first review, it is well known that excessive noise in the workplace has a negative impact on health. The auditory and non-auditory effects of noise on health are evidenced in the scientific literature, and in turn, are taken into account in reports from international organizations (i.e., ILO, WHO, etc.). However, this is a study on the perception of noise by workers. The authors argue that if there is a "convergence" between noise levels above 85 dBA and a negative perception of noise by workers occupying these spaces, it may be the basis for facilitating future hearing conservation programs.
And it is at the moment that I read "workers' perception of occupational noise" when I stop, underline the sentence, prepare a coffee and continue reading very carefully. Why? Because it is not clear to me what the perception of noise means exactly for the authors and what does this perception depend on?
I will try to explain myself. What is the real objective of establishing the perception of noise at the workplace?
- Establish a relationship between perceived (qualitative estimated exposure to noise) and noise measurements (quantitatively measured exposure to noise). In this case, we try to measure the loudness perception or whether the interviewee believes that, the noise dose is high or not.
- Establish the relationship between noise measurements and annoyance (which is linked to the sensitivity of individuals). Noise is one of the most important stressors in the work environment and annoyance is one of the most evident responses to this stressor. Stansfeld and Clark stated that noise annoyance and psychological distress have reciprocal effects on each other.
Therefore, it is absolutely consistent that a person perceives the noise as loud, and even so that it does not cause any inconvenience in their work activities. So figure 1 should split into two figures that describe the perception of loudness and the perception of annoyance. For example, worker 3B said: “It’s loud, especially when the machines gave (sic) that banging, when it gets ready to finish. I’m so use to not wearing them (ear plugs), you get comfortable with the noise”. This probably indicates that the interviewee estimates that the risk to his/her health is more related to the annoyance than to the noise level.
That is why my previous recommendation that I made in the first round of reviews (and that I repeat in this second because it was not taken into account) was a bibliographic review that helps to understand how to measure and what is the perception of noise and of what variables It depends. This would have allowed the authors to reflect on their work.
My recommendation is (again) that a review of the literature is carried out, which will result in a significant improvement of the introduction and discussion sections. This improvement will affect the interpretation of the responses of the interviewees (all this, taking into account that it is no longer possible to vary the interview questions).
Reviewer 2 Report
The data presented represent a lot of work, but I don't believe that you designed your research properly to address your stated aim of investigating the convergence between worker reports and measured sound levels. I suggest reworking the manuscript to avoid focussing on that. As I have said, I do think your interview results will be interesting.
Reviewer 3 Report
The main concern is that most of the comments were not addressed correctly.
